# Broken Access: On the Challenges of Screen Reader Assisted Two-Factor and Passwordless Authentication

## Abstract

In today's technology-driven world, web services have opened up new opportunities for blind and visually impaired people to interact independently. Securing interactions with these services is crucial; however, currently deployed methods of web authentication mainly concentrate on sighted users, overlooking the specific needs of the blind and visually impaired community. In this paper, we address this critical gap by investigating the security and accessibility aspects of these web authentication methods when adopted by blind and visually impaired users. We model web authentication for such users as screen reader assisted authentication and introduce an evaluation framework called Authentication Workflows Accessibility Review and Evaluation (AWARE). Using AWARE, we then systematically assessed popular PC-based and smartphone-based screen readers against different types of deployed web authentication methods, including variants of 2FA and passwordless schemes, to simulate real-world scenarios for blind and visually impaired individuals. We analyzed these screen reader assisted authentication interactions with authentication methods in three settings: using a terminal (PC) with screen readers, a combination of the terminal (PC) and smartphone with screen readers, and smartphones with integrated screen readers. The results of our study underscore significant weaknesses in all of our observed screen reader assisted authentication scenarios for real-life authentication methods. These weaknesses, encompassing specific accessibility issues caused by imprecise screen reader instructions, highlight vulnerability concerning observed scenarios for both real-world and research literature based attacks, including phishing, concurrency, fatigue, cross-service, and shoulder surfing.

Broadly, our AWARE framework can be used by authentication system designers as a precursor to user studies which are typically time-consuming and tedious to perform, independently allowing to unfold security and accessibility problems early which designers can address prior to full-fledged user testing of more isolated issues.

## 1 Introduction

Security researchers and practitioners are continuously working to improve user security through secure and user-friendly authentication methods [9, 15, 73, 75, 88]. However, many studies and real-world methods often overlook the security of accessibility solutions, especially those for authenticating blind and visually impaired users. Currently, 43 million people live with blindness, and 295 million have moderate-to-severe visual impairment [70]. Many of them use assistive technologies [24, 40, 63, 71, 79, 82] to manage online bank accounts, personal accounts, and access web services. Furthermore, unauthorized access to the account of a blind or visually impaired employee within an organization could expose sensitive organizational information, increasing the risk of data breaches [2].

The focal point of our investigation is to evaluate the vulnerability and accessibility of authentication processes involving screen readers, modeling the problem as *screen reader assisted authentication*.

We aim to determine if these processes introduce security risks that could aid attacker's objectives. Single-factor authentication is considered the weakest form of security [17, 25]. Due to accessibility limitations, blind and visually impaired users face challenges in selecting and identifying strong passwords [34, 74], typing passwords on smartphones [4, 59], and also raising security concerns during password entry [41]. Therefore, this research examines secure 2FA, MFA, and passwordless systems (e.g., push-based authentication and Passkeys) using assistive technologies like screen readers [8, 91], the primary medium of interaction with digital services for blind and visually impaired users [26]. Studies have highlighted issues with screen reader accessibility, such as navigation challenges and risks related to headphone usage [1], with a key security concern being potential eavesdropping during user interactions [50, 84], although these studies did not focus on authentication interactions.

Moreover, researchers have explored authentication methods and addressed accessibility issues in web authentication, including locating authentication pages and verifying credentials [27], as well as developing dedicated methods like BraillePassword [5], BendyPass [19], and AudioBlindLogin [46]. However, none of these studies have specifically evaluated widely deployed authentication methods, such as those from Google, Duo, Microsoft, and Authenticators from different vendors for various platforms (e.g., smartphone, desktop), concerning both security and accessibility aspects. This gap includes understanding interaction dynamics with screen readers and their response to attacks targeting authentication methods. Expanding research in this area could offer valuable insights for improving the security and accessibility of authentication processes for individuals with blindness or visual impairments, which is the focus of our study.

To conduct our study, we designed the Authentication Workflows Accessibility Review and Evaluation (AWARE) framework, focusing on analyzing interactions between screen readers and authentication methods. The AWARE framework is designed to evaluate the security and accessibility of authentication methods for blind and visually impaired users who rely on screen readers. It works through a semi-automated process that records and analyzes screen reader interactions with various authentication workflows, using the speech-to-text conversion of the traversed authentication workflow and comparing it with the original text to assess how well instructions are conveyed. It also tests responses to simulated attacks, such as phishing, shoulder surfing, concurrent login, and notification fatigue. The framework helps identify issues early, making it a cost-effective tool for developers to improve the security and usability of authentication systems before conducting full-scale user studies.

We began with a preliminary analysis to select appropriate screen readers and authentication methods, covering a range of real-life scenarios. In the evaluation phase, we used the AWARE framework to perform authentication with the selected methods, utilizing screen readers such as JAWS and VoiceOver. This phase included generating attacks drawn from both real-world scenarios and research literature to assess the authentication methods thoroughly.

**Our Contributions.** Key contributions are highlighted below:

- ***The notion of screen reader assisted authentication.*** We model the studied problem as screen reader assisted authentication and systematically examine security and accessibility challenges in real-life authentication methods, including several 2FA and passwordless variants such as OTP, push notifications, FIDO, and phone calls. Unlike prior research, we assessed these methods from the perspective of blind and visually impaired users. We aimed to understand how these users interact with authentication methods in computers and smartphones.

- ***A qualitative and quantitative framework and methodology for security and accessibility assessment.*** We present the AWARE framework and methodology to evaluate diverse real-life authentication methods using various screen reader environments. We selected different 2 and MFA variants for assessment with chosen terminal (PC)-based and smartphone-based screen readers in three settings: using a terminal (PC) with screen readers, a combination of terminal (PC) and smartphone with screen readers, and smartphones with integrated screen readers. These selections and usage settings cover a wide range of settings to accommodate user preferences. By conducting different attacks including phishing, concurrency [72], fatigue [51], cross-service [62], shoulder surfing, and downgrading [87] attacks in screen reader assisted authentication scenarios, we aim to understand the security and accessibility challenges for blind and visually impaired individuals. *Our AWARE framework and methodology can be used as a precursor to user studies, which can be costly and time-consuming especially due to the need to recruit study subjects who are blind or visually impaired.* It can help isolate critical accessibility and security problems that the designers of the authentication system can focus on addressing prior to user testing or deployment.

- ***A systematic evaluation of web authentication for targeted users.*** Following our AWARE framework and methodology, we evaluated twelve different types of real-life 2FA methods using six different screen readers on both terminals (PCs) and smartphones, identifying critical vulnerabilities and accessibility challenges. The accessibility issue arises from a stark difference in comprehensibility, where screen readers achieve a comprehensibility of over 74% for general text but drop significantly for authentication instructions, with only 22 out of 33 cross-settings involving screen readers and authentication methods reaching less than 50% comprehensibility. Vulnerability challenges include the screen reader's inability to pronounce and identify phishing links, which may enable victims to inadvertently share OTP codes over the attacker's phishing channel. We observed exploitation against push-based 2FA across different attacks and vulnerabilities against FIDO-MFA and passwordless (Passkeys) for numerous attacks including downgrading and cross-service attacks. These findings indicate that screen reader-assisted users may face higher vulnerability in real-life authentication methods than those without screen readers.

**Demonstrations:** We demonstrate some of our key findings at: https://sites.google.com/view/secure-auth-for-blind-user/home

## 2 Related Work

**Security Exploitation of Screen readers.** Researchers have studied screen readers, which allow visually impaired and blind users to access the internet and smart devices. Inan et al. [48] surveyed 20 individuals who used screen readers for internet browsing. Their research revealed security issues, including misleading links, spam emails, and poorly designed CAPTCHA verifications on web pages. Hasan and Gjøsæter [42] identified research gaps and accessibility problems in interactive maps, proposing potential solutions. Hayes et al. [43] conducted a study in which participants expressed security concerns while using screen readers at home, work, and public places. Ambore et al. [6] surveyed five participants to assess security-accessibility trade-offs in mobile financial services for visually impaired users. Their findings revealed usability and security issues, including the need to audibly express passwords. While these studies highlighted security concerns associated with screen readers, they did not specifically investigate the vulnerability and accessibility of authentication methods for blind and visually impaired users who rely on screen readers.

**Security and Usability Concerns of Authentication Technology for Blind and Visually Impaired Users.** Researchers also reported on the usability aspects of authentication systems for blind and visually impaired users. Dosono et al. [26] conducted a contextual survey on blind and visually impaired individuals' authentication experiences across computers, smartphones, and websites. Their findings indicated significant authentication delays and confusing login challenges. In a literature review by Andrew et al. [7], accessible authentication mechanisms for individuals with various impairments were analyzed, including individuals with blindness and low vision. The review highlighted the lack of thorough usability assessments in prior research and the limitation of small sample sizes that did not adequately represent the target audience. Saxena and Watt [78] discussed authentication technologies for blind and visually impaired users a decade ago, focusing on secure user and device authentication. Their paper proposed research challenges and directions for authentication technologies. Faustino and Girouard [20] conducted a survey to understand the security and usability challenges of authentication methods used on mobile devices and highlighted the usability challenges of blind and visually impaired users. The results highlighted a preference for easy-to-use authentication methods, such as fingerprint recognition, over other methods like PIN-based authentication. Unlike our work, these studies did not focus on the security challenges of widely used real-life authentication methods from the perspective of blind and visually impaired users.

In Appendix Section 8.1, more related studies are discussed in detail.

## 3 Preliminaries

To achieve our research objective, we methodically selected screen readers and authentication methods, as described in this section.

### 3.1 Selection of Screen Readers

In choosing screen readers, we aimed to consider a wide range of fully-featured options for both mobile and terminal (PC) platforms, focusing on their authentication assistance capabilities. Factors like numbers of downloads and insights from studies, particularly "Screen Reader User Survey Number 9" by WebAIM [91], influenced our decision. This study holds significant relevance to our research, with 92.30% of respondents identifying as disabled.

Based on the mentioned survey and our investigation, we have identified user preferences for terminal (PC)-based options such as JAWS, NVDA, Dolphin, and ChromeVox. It is worth noting that

ChromeVox, a browser-based screen reader, was included in our list as a terminal (PC)-based screen reader and has over $200,000$ users, as indicated in the Chrome Web Store [36].

Participants of the mentioned study indicated VoiceOver and TalkBack, the default screen readers for iPhone and Android. We also attempted to include more smartphone-based screen readers by searching the App Store and Google Play. However, after installing the apps from these searches, we found that most were designed for reading web pages, PDFs, or for object identification. Examples include Speechify Text to Speech Voice, NaturalReader - Text to Speech, Audify read aloud web browser, and TextGrabber Scan and Translate. Unfortunately, we did not find any smartphone based screen readers that can perform full interaction including authentication apart from VoiceOver and Talkback for our study. Appendix Table 4 lists the selected screen readers for both PC and smartphones.

### 3.2 Selection of Authentication Methods

For selecting authentication methods, we ensured comprehensive coverage of various real-life methods, as detailed in Appendix Table 5, chosen based on their popularity (see Appendix Table 6). Methods utilizing a one-time password mechanism, such as text messages, phone calls, and authenticators, were labeled as "One-time password (OTP-2FA)", while push notification methods were categorized as "Push-2FA". Additionally, we tested Google's Titan Security Key [37], categorized as "FIDO-MFA", as it requires inserting the Fast Identity Online (FIDO) USB key into the device and then putting the user's fingerprint on the key for authentication. Duo's phone calls instruct users to press a specific number key in the keypad for authentication, it is denoted as "Phone call-2FA".

Authenticators generate OTPs for a certain duration for registered accounts to perform authentication. To investigate authentication within the terminal (PC), we selected authenticators in the form of desktop applications and browser extensions, including *Authenticator* (browser extension), *Twilio Authy* (desktop application) which has popular mobile version applications, *WinAuth* (desktop application) [3], and *GAuth* (browser extension). Some authentication methods are compatible with both PC and smartphone platforms.

We chose Duo (OTP-2FA, Push-2FA, Phone Call-2FA) based on the number of downloads, and user priority [30, 33]. Additionally, we included authentication methods provided by Google (OTP-2FA, Push-2FA, FIDO-MFA) in our study based on user preferences [16]. Microsoft's select-confirm Push-2FA is a passwordless approach that requires users to authenticate by confirming a number displayed in the login window. This confirmation is achieved by selecting the corresponding number from options presented in the push notification and unlocking the phone with face recognition, fingerprint, or PIN.

## 4 Our Aware Framework and Study Methodology

In this section, we present the Authentication Workflows Accessibility Review and Evaluation (AWARE) framework, a simple qualitative and quantitative method for evaluating the security and accessibility of screen reader-assisted web authentication. As one of the first works in this under-studied research domain, the AWARE framework is designed to identify critical issues in screen reader-assisted web (2FA/passwordless) authentication and serves as a semi-automated, inexpensive pre-study tool to highlight major concerns that can be investigated further with future user studies. Our framework highlights

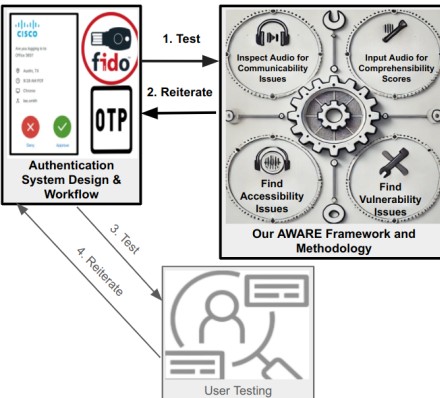

**Figure 1: AWARE framework and methodology as a precursor to traditional user study.**

key concerns that designers should prioritize, rather than relying solely on user studies, as organizing such studies with this group is challenging. Designers can iteratively update and re-evaluate their solutions based on the AWARE framework's findings.

### 4.1 Process of Evaluating Screen Readers

The framework allows us to assess the communicability of the screen readers, which refers to the screen reader's ability to convey instructions without any inconsistencies, by pinpointing instances where critical authentication details (such as OTPs and essential instructions) were inadequately communicated. These issues in communication could hinder blind and visually impaired users' ability to complete the authentication process, as elaborated in Section 5.1.

The framework additionally allows us to assess screen readers' ability to communicate information to the visually impaired users within web pages (for web apps) or application instances (for desktop and smartphone applications), which we termed as "comprehensibility". Comprehensibility, in this context, pertains to the clarity and relevance of the conveyed information [14] for proper authentication. Furthermore, the framework allows us to conduct a comparative analysis of comprehensibility between authentication system workflows and general-purpose web pages, such as news articles, to determine if screen readers effectively convey information for authentication system workflow compared to these broader content types (serving as a baseline for evaluation). The reason for low comprehensibility indicates the difficulty faced by screen readers because of inaccessible authentication interfaces such as the structure of presented content (e.g., maintaining heading label), alternative text for images, explain buttons, and text box adequately.

To evaluate comprehensibility, the framework takes the input of recorded output of screen readers for a news article as a general text and the authentication flow of each authentication method as authentication instruction texts in the default reading speed, which many new (non-expert) blind users prefer [18]. These recorded audio files are then converted to text using the IBM Watson speech-to-text engine [47] automatically. The generated text was compared to the original text to measure comprehensibility automatically through similarity and dissimilarity, indicating clarity.

The framework chose the speech-to-text engine due to its significant advancements and high accuracy in speech recognition technology. In certain cases, it has even outperformed professional human

transcribers [92]. To compute the similarity between the original and generated text, it utilized the pysimilar library, which is a Python library that calculates cosine similarity [52]. Pysimilar utilizes the TF-IDF vectorizer to transform the text into vectors, converting them into arrays of numbers. Subsequently, cosine similarity computation is performed. Term frequency - Inverse document frequency (TF-IDF) serves as a text vectorizer that transforms the text into a usable vector [60]. Using TF-IDF vectors, cosine similarity proved most effective in identifying similarities among short texts [81].

The collection process for recording authentication workflows is assisted by humans, while the evaluation process is automated, as explained above. Therefore, we describe our AWARE framework and methodology as semi-automated and a precursor to user testing. In traditional user testing-based evaluations of workflows, developers need to run the base system and all authentication flows during user study. In contrast, our AWARE framework and methodology allow for testing authentication flows without user tests; instead, the developer tests them as they would test web accessibility. Although it involves some manual effort, this approach helps identify accessibility issues and security vulnerabilities earlier in the process. Developers can then iterate on their system to address these issues and may run the authentication workflow using our AWARE framework and methodology again. Once the developer is satisfied with the results, and if desired, the final design can be subjected to small-scale user testing to uncover any new issues that our AWARE framework and methodology might have missed. This process is illustrated in Fig. 1. Therefore, our approach is not intended to replace user testing but to reduce the cost and frequency of running user tests on entire workflows. User studies can be designed by the findings from our methodological evaluation and conducted after achieving a more refined design.

## 4.2 Evaluating Screen Reader Assisted 2FA & MFA

To evaluate the vulnerability and accessibility of 2FA and MFA methods, we employed the AWARE framework and methodology with screen readers. Additionally, many blind and visually impaired users may use headphones and a screen curtain to keep themselves secure from shoulder surfers while performing authentication [12, 93]. Based on the WebAIM screen reader user survey number 8 [90], 41.3% of users utilize terminal (PC)-based screen readers, 9.5% use smartphone-based screen readers, and 49.2% use both simultaneously. Therefore, the framework structured the test settings accordingly, as outlined in Appendix Table 7 and explained below. While we described our selected screen readers and authentication methods for each setting, designers can choose their own screen readers or authentication methods for their evaluations.

**Terminal (PC)-based Screen Readers.** Terminal (PC)-based screen readers, including JAWS, NVDA, Dolphin, and ChromeVox, were used to assess a range of terminal (PC)-supported authentication methods. The tested methods include Google's FIDO-MFA and OTP-2FA via authenticators (e.g., authentication codes from Authy, GAuth, and WinAuth). Methods not supported in terminal (PC)-based scenarios, like Duo (specific to smartphones), were excluded.

**Smartphone-based Screen Readers.** We used smartphone-based screen readers, Talkback (Android) and VoiceOver (iPhone), to evaluate authentication methods. However, some terminal (PC)-based authenticator software like GAuth and WinAuth is not supported on smartphones, so we focused our testing on the remaining methods.

**Simultaneous use of Terminal (PC)-based and Smartphone-based Screen Readers.** We set up both a terminal (PC) and smartphones with screen readers to evaluate authentication methods. Our focus was on understanding how these methods respond to screen readers simultaneously on different devices. From Appendix Table 8, it is evident that there are a total of 8 possible combinations involving 2 smartphones and 4 PC-based screen reader configurations. However, upon observation, we noted that changing the PC-based screen reader (e.g., JAWS, NVDA) with a particular smartphone-based screen reader (e.g., VoiceOver or Talkback) did not yield significant changes or effects while evaluating PC and Smartphone concurrent use combinations. Here, the choice of smartphone-based screen reader had a more significant impact when compared to altering the screen reader on a PC-based terminal. As a result, we narrowed down our testing to only 2 combinations (JAWS-VoiceOver, NVDA-Talkback) in the PC and Smartphone concurrent use combinations.

## 4.3 Evaluating Existing Threats Against Screen Reader Assisted Authentication Scenario

**Threat 0: Remotely Fingerprinting Visually Impaired Users.** Attackers may target blind and visually impaired users, viewing them as vulnerable. Detecting these users could be the first step in exploiting further vulnerabilities. Momotaz et al. [66] highlight the use of extensions to enhance accessibility, which can be detected via browser fingerprinting techniques [77, 83, 85, 86] to identify blind or visually impaired users. We conducted a technical inspection based on the noted studies and accurately detected ChromeVox and other listed extensions. Details of this attack are in Appendix Section 8.2.

**Threat 1: Phishing.** Fraudulent requests often aim to deceive victims into revealing credentials via phishing [68]. Blind and visually impaired users rely on screen readers, which often struggle to convey subtle differences in URLs because they pronounce top-level domains as whole words, complicating the detection of phishing attempts. We assessed slightly altered top-level domains, such as *bankofamerica* vs. *bankoffamerica* and *wellsfargo* vs. *wellssfargo*, using the same methods described in Section 4.1. Our study found that screen readers pronounced these pairs similarly, and similarity scores showed a 100% match (Appendix Fig. 2). While users can manually check links character-by-character using screen readers, this method is time-consuming, requires extra keystrokes, and attackers may exploit this by creating long or complex URLs, further obscuring detection. Additionally, Lau and Peterson [57] highlighted that screen readers struggle to properly read phishing warnings generated by browsers. Attackers can exploit these limitations, using similar-looking URLs and unreadable phishing warnings to deceive visually impaired users and breach security (see Appendix Fig 3).

**Threat 2: Concurrent Login.** In this scenario, an attacker initiates a login session simultaneously with the victim, deceiving the victim into accepting the attacker's approach, mistakenly believing it is their own [56, 62, 72]. An attacker can learn probable login times by monitoring the victim's browsing sessions, exploiting contextual information like peak usage times (e.g., start of a workday, after breaks, observing the victim's login behavior in physical proximity, or using social engineering techniques such as prompting the victim

to log in during a fake troubleshooting session [53]). In this research, we specifically observed this vulnerability in Push-2FA variants from the perspective of blind and visually impaired users. Jay Prakash et al. simulated this attack on notification-based 2FA with 75 user-attacker pairs, discovering only 5% expressed doubts about the attack [72].

When testing Google Push with VoiceOver and Talkback, we found that the attacker's concurrent notification overrides the legitimate one. Although the notification page shows device and city details, success depends on the victim accepting the attacker's push, either assuming it's legitimate or ignoring the details. Attackers can also initiate logins from the same device and city as the victim, as these are often generic. Duo Push is particularly vulnerable, as it overrides the victim's notification and lacks attacker device details, making it nearly identical. Microsoft select-confirm appears more secure, requiring users to confirm a number and unlock the phone via fingerprint, face, or PIN. However, Mahdad et al. [62] describe an attack where the attacker blocks the user's request and manipulates the authentication to present malicious options. Screen reader assisted users will not have any instruction about this attack and may mistakenly accept the attack session, assuming it to be legitimate.

**Threat 3: Notification Fatigue.** Push notification fatigue, known as MFA spamming, involves continuously sending prompts to the user until they accept it, become mentally exhausted, or the attacker ceases the attack [51]. For our observation, we fatigued on each targeted authentication method from the perspective of blind and visually impaired users in screen reader-assisted scenarios by continuously generating push notifications at fixed $t$ time intervals, shown in Appendix Fig. 4. Our observations found that Google and Duo's push-based methods are vulnerable.

Authentication via Google Push notification is highly vulnerable with VoiceOver and Talkback. In VoiceOver, earlier notifications remain at the top, and denying the attacker's push notification redirects the victim to change passwords. After pressing "change password", the victim might mistakenly accept the next (attacker) push notification, assuming it is a password change process. In Talkback, the attacker's recent notification overrides the previous one, potentially confusing the victim and leading to accepting malicious push during password changes. Microsoft select-confirm is also vulnerable to push fatigue. After declining an attacker's push, the victim receives simple "approve" and "deny" options rather than select-confirm nature, which attackers can repeatedly trigger, causing frustration and potentially leading to acceptance out of exhaustion. Duo Push vulnerability depends on admin settings that limit attempts, where repeated notifications can lock the account.

**Threat 4: Shoulder-Surfing.** Attackers can obtain sensitive information through shoulder surfing, often without the user's awareness. This threat is heightened by hidden cameras and is especially risky for visually impaired users, who may struggle to detect an attacker's presence [44] or surveillance devices [78]. Our study found that screen readers read credentials aloud during authentication. While users may use headphones or screen curtains for protection on a single device, when using both a terminal (PC) and smartphone concurrently (e.g., receiving OTP on a smartphone while logging in from a PC), one device often remains insecure. Additionally, conflicts arise when phone calls for OTP and screen reader instructions overlap, creating further insecurity (CBI) as discussed in Section 5.

In our scenario, the attacker, in proximity or using monitoring devices, initiates authentication simultaneously with the victim. When the victim clicks the login button, a legitimate OTP is generated. However, if the attacker triggers a "forgot password" request at the same time, the OTP is sent to the victim, who assumes it is for their request. The screen reader then reads the OTP aloud, allowing the attacker to hear it and use it to complete the "forgot password" request, gaining unauthorized access (see Appendix Fig. 6). The behaviors of different OTPs during such attacks are detailed in Section 5.

**Threat 5: FIDO-Specific Threats.** We evaluated the impact of FIDO-MFA on visually impaired users with screen readers, analyzing attacks from the FIDO specification [13] and creating additional scenarios to identify vulnerabilities. Since devices like Google Titan and Yubikey use the same FIDO2 protocols (WebAuthn and CTAP), we focused the evaluation by assessing a single representative device.

*5.1: Display Overlay Attack.* This attack disrupts transactions by presenting false information over legitimate content. We found that JAWS, Dolphin, and ChromeVox were vulnerable, reading the false information, while NVDA bypassed and read the correct data.

*5.2: Phishing.* Phishing challenges blind users as screen readers make phishing links sound similar to legitimate ones (see Section 4.3, Threat 1). Attackers could exploit this to steal credentials and downgrade to weaker 2FA like OTP (see Section 4.3, Threat 5.6).

*5.3: Mis-Authentication and 5.4: Mis-Registration.* These client-side threats occur when users register keys or authenticate on phishing sites, leading to credential compromise. Section 4.3 (Threat 1) described the risk of phishing for blind and visually impaired users.

*5.5: Cross-Service Attack.* In a cross-service attack, the attacker manipulates a user's possession factor device, tricking them into approving authentication for a different service, as explained in Appendix Fig. 5. ChromeVox cannot read Windows security dialogue boxes/ prompts, leaving visually impaired users vulnerable in a FIDO cross-service attack [56]. The Dolphin can read security messages but fails when drawing a fake overlay over legitimate security boxes. JAWS reads the security box but does not read the service or browser name. NVDA performs best by reading the service name even through overlays, and preventing the attack.

*5.6: Downgrading.* Ulqinaku et al. used social engineering to downgrade FIDO, replacing it with weaker two-factor authentication [87]. Attackers exploit screen reader limitations to detect and express phishing links to perform real-time phishing in downgrading attacks. No screen reader can express and detect phishing links instead of expressing them as legitimate, as explained in Section 4.3 (Threat 1). Additionally, screen readers can not detect fake prompts generated by attackers to achieve weaker OTP-2FA code. Hence this attack is marked as vulnerable to screen readers assisted users.

## 5 Results and Findings

### 5.1 Findings on Screen Reader Output

We identified several issues in effectively communicating authentication information (e.g., OTPs, push notifications) and instructions to blind and visually impaired users by utilizing our AWARE framework. As outlined in Section 4.1, these are referred to as *communicability* and *comprehensibility* issues.

Table 1 highlights observed communicability issues that disrupt the transmission of critical authentication information. These issues

cause difficulties in properly completing the authentication workflow, thereby introducing new vulnerabilities for blind and visually impaired users. To represent these issues in Table 1, we have introduced several notations, which we elaborate on below.

**Conflict Between Instructions (CBI).** CBI happens when screen readers deliver critical instructions (e.g., navigating the authentication interface) that conflict with another authentication step, such as receiving an OTP via phone call. This simultaneous communication can confuse users, leading to important information being missed. Additionally, we observed that this conflict can cause headphones to disconnect and the loudspeaker to activate, increasing the risk of shoulder-surfing attackers overhearing sensitive information.

**Table 1: Screen reader's communicability on critical authentication instructions.**

| 2FA/MFA Methods | Terminal (PC) based screen readers | | | | Smartphone based screen readers | |
|---|---|---|---|---|---|---|
| | JAWS | NVDA | Dolphin | ChromeVox | VoiceOver | Talkback |
| Google OTP (text message) | N/A | N/A | N/A | N/A | | NPO |
| Google OTP (call) | N/A | N/A | N/A | N/A | CBI | CBI |
| Duo text message | N/A | N/A | N/A | N/A | NPO | NPO |
| Google authenticator | | | | | NPO | NPO |
| GAuth authenticator | UCO | UCO | UCO | UCO | N/A | N/A |
| WinAuth authenticator | UCO | UCO | UCO | UCEOB | N/A | N/A |
| Authy authenticator | | | NPO | UCEOB | NPO | NPO |
| FIDO (Titan Security Key) | UCSP | | UCSP | UCEOB | | |
| Duo, call me | N/A | N/A | N/A | N/A | CBI | CBI |

N/A means unfeasible/unsupported; empty field: can explain critical information

**Numeric Pronunciation of OTP (NPO).** We observed that some screen readers pronounce OTPs as numeric values (e.g., "one thousand two hundred thirty-four" for "1234") instead of articulating digit-by-digit. This issue, labeled *Numeric Pronunciation of OTP (NPO)*, becomes more problematic with longer OTPs.

**Unable to Communicate OTP (UCO).** We have observed instances where some screen readers are unable to pronounce OTPs because of the authenticator's interface such as not allowing the screen to read the OTP, leading to the failure to convey this critical authentication information to visually impaired users, as listed in Table 1.

**Unable to Communicate Security Prompts (UCSP).** In some authentication systems, such as FIDO key-based authentication, users encounter security prompts (confirmation messages containing service name and browser name) in their workflow, typically generated by the operating system (e.g., Windows Security). We have observed that some screen readers fail to read this critical authentication information altogether, while others stop pronouncing instructions (e.g., "insert your security key") to pronounce the contents of the prompts.

**Unable to Communicate Elements Outside Browser (UCEOB).** In some authentication workflows (e.g., FIDO-MFA), critical information is presented outside the browser (e.g., Windows Security messages). Browser-based screen readers like ChromeVox cannot convey this critical information as it falls outside their scope.

To complement the communicability analysis, we measured the comprehensibility of screen readers during the authentication workflow using the IBM Watson speech-to-text engine (see Section 4.1).

Our focus was on whether screen readers can convey all authentication information (e.g., locating password/User ID fields, pronouncing instructions, communicating OTPs). Results, shown in Table 2, revealed low comprehensibility for GAuth via JAWS, NVDA, Dolphin, and OTPs via VoiceOver and Talkback (under 50%), while Dolphin via FIDO showed higher percentages (75.47%). Comprehensibility percentages reflect how well screen readers cover written content, including both essential elements (e.g., buttons, OTPs) and non-essential details (like taglines or service information) within the authentication interface. However, higher percentages do not necessarily indicate better overall communication given the existing communicability issues, as discussed in Table 1.

**Table 2: Comprehensibility of authentication workflows.**

| 2FA/MFA Methods | Terminal (PC) based screen readers | | | | Smartphone based screen readers | |
|---|---|---|---|---|---|---|
| | JAWS | NVDA | Dolphin | ChromeVox | VoiceOver | Talkback |
| Google OTP (text message) | N/A | N/A | N/A | N/A | 20.58% | 20.81% |
| Google OTP (call) | N/A | N/A | N/A | N/A | | |
| Duo text message | N/A | N/A | N/A | N/A | 44.84% | 27.38% |
| Google authenticator | 47.43% | 24.96% | 2.20% | 56.89% | 16.82% | 26.40% |
| GAuth authenticator | 4.45% | 16.33% | 11.30% | 65.72% | N/A | N/A |
| WinAuth authenticator | 12.08% | 57.45% | 54.08% | | N/A | N/A |
| Authy authenticator | 46.16% | 69.69% | 40.26% | | 37.32% | 34.48% |
| Microsoft select-confirm | N/A | N/A | N/A | N/A | 67.53% | 41.28% |
| Duo push | N/A | N/A | N/A | N/A | 44.84% | 31.43% |
| Google push | N/A | N/A | N/A | N/A | 58.93% | 45.47% |
| FIDO (Titan Security Key) | 32.74% | 67.99% | 75.47% | | 85.82% | 86.18% |
| Duo, call me | N/A | N/A | N/A | N/A | | |

N/A indicates unfeasible, unsupported, and empty means mentioned in Table 1

To compare the comprehensibility percentages with other general-purpose web pages, we conducted a similar assessment on a general news article. The results revealed a consistent level of good comprehensibility, with percentages ranging from 74.63% to 89.83%, as shown in Appendix Table 9. Unlike well-structured text in news articles, authentication interfaces often contain critical visual cues (e.g., "touch the key", or OTP communication) that screen readers cannot interpret, resulting in lower scores and increased difficulty.

## 5.2 Vulnerability and Accessibility Analysis

Our observation of screen reader assisted 2FA and MFA methods via AWARE framework and methodology suggest numerous vulnerabilities, as depicted in Table 3. Appendix Table 10 displays accessibility metrics, including feasibility, exceeding verification time, and conflict between instructions (CBI). These metrics, along with the communicability and comprehensibility of instructions from Table 1 and 2, were used to evaluate the accessibility of various methods.

**Feasibility** refers to the successful use of an authentication method, classified as feasible (no communication issues), partially feasible (e.g., confusing OTP pronunciation), or not feasible (failure to provide crucial instructions). Our feasibility determination method focuses on technical communication problems, excluding human factors (e.g., habituation, intuition). **Exceeding verification time** indicates prolonged authentication due to authentication methods interface inconsistencies, causing improper screen reader instructions and delays. Table 1 outlines communicability issues, while Table 2

**Table 3: Vulnerability of authentication methods to attacks.**

| | | Push-2FA | | | | One Time Password-2FA | | | | | | |
|---|---|---|---|---|---|---|---|---|---|---|---|---|
| | | Microsoft, select-confirm | Duo, Push | Google, Push | Duo, call me | Duo, text message | Google, text message | Google, call | GAuth Authenticator | WinAuth Authenticator | Authy Authenticator | Google Authenticator |
| **JAWS** | Shoulder Surfing | | | | | | | | ○ | | ○ | ○ |
| | Phishing | | | | | | | | ● | | ◐ | ● |
| **NVDA** | Shoulder Surfing | | | | | | | | ○ | ○ | ○ | ○ |
| | Phishing | | | | | | | | ● | | ◐ | ◐ |
| **Dolphin** | Shoulder Surfing | | | | | | | | ○ | | ○ | ○ |
| | Phishing | | | | | | | | ● | | ◐ | ● |
| **ChromeVox** | Shoulder Surfing | | | | | | | | ○ | | | ○ |
| | Phishing | | | | | | | | ● | | | ◐ |
| **VoiceOver** | Concurrency Attack | ● | ● | ● | | | | | | | | |
| | Shoulder Surfing | ○ | ○ | ○ | | ○ | ○ | ● | | | ○ | ○ |
| | Phishing | ○ | ○ | ○ | | ◐ | ● | | | | ● | ◐ |
| | Fatigue Attack | ● | ◐ | ● | | | | | | | | |
| **Talkback** | Concurrency Attack | ● | ● | ● | | | | | | | | |
| | Shoulder Surfing | ○ | ○ | ○ | | ○ | ○ | ● | | | ○ | ○ |
| | Phishing | ○ | ○ | ○ | | ◐ | ◐ | | | | ◐ | ◐ |
| | Fatigue Attack | ● | ◐ | ● | | | | | | | | |
| **JAWS with VoiceOver** | Concurrency Attack | ● | ● | ● | | | | | | | | |
| | Shoulder Surfing | ○ | ○ | ○ | | ● | ● | ◐ | ● | | ● | ● |
| | Phishing | ○ | ○ | ○ | ● | ◐ | ● | | | | ● | ◐ |
| | Fatigue Attack | ● | ◐ | ● | | | | | | | | |
| **NVDA with Talkback** | Concurrency Attack | ● | ◐ | ● | | | | | | | | |
| | Shoulder Surfing | ○ | ○ | ○ | | ● | ● | ◐ | ● | | ● | ● |
| | Phishing | ○ | ○ | ○ | ● | ◐ | ◐ | | | | ● | ● |
| | Fatigue Attack | ● | ◐ | ● | | | | | | | | |

● authentication is vulnerable. ◐ fifty-fifty vulnerability. ○ authentication is not vulnerable. Empty means the attack is not relevant or was not tested.

shows how well screen readers guide users during authentication. Unclear instructions can lead to confusion and security risks.

**Terminal (PC)-based Screen Readers.** In this setting, we tested only terminal (PC)-supported authentication methods, so metrics for other methods are excluded in Table 3 and Appendix Table 10. Vulnerability for concurrent login and notification fatigue is specific to push-based authentication methods and is not represented in Table 3 for other authentication methods. None of the methods are susceptible to shoulder surfers under this setting, as victims may use headphones and screen curtains during the authentication process.

OTP by GAuth is feasible for all terminal (PC)-based screen readers, despite the 60-second validity of the authentication code. In contrast, OTP by WinAuth is infeasible as screen readers are unable to communicate the code (UCO). WinAuth displays OTPs as asterisks, requiring a mouse click to reveal them, but even then, screen readers cannot vocalize the code to the user. Therefore, WinAuth is categorized as not applicable to phishing, while GAuth remains vulnerable (Section 4.3, Threat 1). Success in phishing depends on the feasibility of authentication methods, as the user's ability to input the code is required. Additionally, ChromeVox cannot provide instructions for WinAuth due to its inability to communicate elements outside the browser (UCEOB).

OTP by Twilio Authy, a desktop app, is inaccessible to ChromeVox due to its inability to read outside the browser (see Section 5.1). It is partially vulnerable to phishing for the terminal (PC)-based screen readers (except ChromeVox) as it provides only partial feasibility and meaningful output. OTPs by Authy remain valid for 60-second, and transitioning between login and authentication windows takes

time, hindering clear instructions from screen readers. NVDA struggles to read button names, while Dolphin pronounces OTPs in two parts (e.g., "one hundred twenty-three, four hundred fifty-six" for "123456") (see Section 5.1 (NPO)).

OTP by Google Authenticator is vulnerable to phishing with JAWS and Dolphin. NVDA and ChromeVox face the numeric pronunciation of OTP (NPO) and lack an easy method to copy and input the code, making phishing vulnerability partial. The code disappears in 30 seconds and expires in 60 seconds, complicating the process.

In Table 3, FIDO (Titan Security Key) feasibility is categorized as partial for screen readers due to imprecise instructions identified in Table 2 and the inability to communicate security prompts (UCSP) for key insertion (Section 5.1). Dolphin and ChromeVox lack clear timing instructions for key insertion, though experienced users may authenticate through assumptions. However, this raises the risk of users accepting an attacker's approach, explained in Section 4.3 (Threat 5) and shown in Appendix Table 11.

**Smartphone-based Screen Readers.** We evaluated all four smartphone supported authentication methods shown in Appendix Table 5. Push-based methods like Microsoft select-confirm, Duo Push, and Google Push offer feasibility and defense against shoulder surfing and phishing but are vulnerable to concurrent login and notification fatigue with VoiceOver and Talkback. In concurrent login, attacker notifications override legitimate push notifications, causing the screen reader to read the recent (which can be attacker-generated) notification (see Section 4.3 (Threat 2)). In Section 4.3 (Threat 3), we discussed notification fatigue. Only Duo Push prevents the attacker from generating continuous push notifications, making it partially vulnerable due to administrators setting a threshold for concurrent push notifications to block attackers, as indicated in Table 3.

The Duo phone call method is infeasible due to conflict between instructions (CBI), as explained in Section 5.1. Duo OTP via text is partially vulnerable to phishing for smartphone screen readers due to numeric OTP pronunciations (NPO), making it difficult for blind users to remember a seven-digit code. OTP via Google Text Message exhibits the same vulnerability as the Duo Text Message method. Google OTP is vulnerable to phishing with VoiceOver but only partially vulnerable with Talkback, as VoiceOver does not convey the numeric pronunciation of OTP (NPO). Google OTP via phone call is also infeasible due to conflict, and it disconnects headphones, switching to speaker mode, increasing the risk of shoulder surfing.

OTP by Twilio Authy (mobile app) shows vulnerabilities with smartphone-based screen readers. VoiceOver allows easy code copying, increasing feasibility and phishing vulnerability, while Talkback struggles with copying, resulting in partial vulnerability to phishing due to limited feasibility. Similarly, Google Authenticator (app) pronounces the code numerically (NPO), making both VoiceOver and Talkback partially feasible and partially vulnerable to phishing.

Google FIDO-MFA works well with VoiceOver, but Talkback provides inadequate instructions. FIDO's threat is detailed in Section 4.3 (Threat 5). However, FIDO has convenient usability.

**Simultaneous Use of both Terminal (PC) and Smartphone Screen Readers.** In these settings, all combinations are marked as "extremely vulnerable" due to the limitation of the user using headphones for both devices simultaneously. "Extremely vulnerable" perhaps indicates an unavoidable compromise, while "partially vulnerable" suggests potential attacks or defenses in specific situations.

The Duo and Google push methods are feasible but lack clear authentication instructions, making them vulnerable to concurrent login (Section 4.3, Threat 2) and notification fatigue (Threat 3). However, Duo Push is only partially vulnerable to fatigue with admin-controlled attempt limits. Microsoft's select-confirm method is feasible in all combinations, but terminal (PC)-based screen readers take longer to read the value from the login window to select in the smartphone, and smartphone-based readers need extra time to select the number due to the interface. These delays expose conflicts between terminal (PC) and smartphone screen readers (CBI). The method is vulnerable to concurrent login (Threat 2) and notification fatigue (Threat 3), where it simplifies to "approve" and "deny" buttons during fatigue attacks.

The Duo Phone Call is not feasible due to instruction conflicts (CBI), as explained in Section 5.1. OTP methods are vulnerable to shoulder surfing across all screen reader combinations in this setting due to an unprotected device without headphones, allowing unauthorized access (Section 4.3, Threat 4). Duo Text Message has partial feasibility, with numeric pronunciation (NPO) and slow switching between message and login windows, making it partially vulnerable to phishing. Google Phone Call for OTP introduces conflicts (CBI) that trigger the phone's speaker, potentially exposing the OTP.

This setting shows similar accessibility and vulnerability for GAuth (extension) and WinAuth (desktop app) as terminal (PC)-based screen readers due to identical testing procedures. For OTP via Google and Authy, the desktop versions behave like terminal (PC)-based settings, while the smartphone apps simplify authentication by allowing direct code input. However, this simplicity increases vulnerability to shoulder surfing, as codes are spoken aloud, and the method's feasibility makes it vulnerable to phishing.

In this setting, we tested FIDO-MFA for terminal (PC) authentication. Combinations showed similar partial feasibility, as screen readers are unable to communicate security prompts (UCSP), as detailed in Section 5.1 with instructions to touch the key for authentication. The Dolphin also lacked key placement instructions. Detailed FIDO vulnerabilities are discussed in Section 4.3 (Threat 5).

## 6 Discussion and Further Insights

**Limitations and Potential Mitigations.** While our study offers valuable insights into the accessibility and security vulnerabilities associated with screen readers and their responses to various authentication methods, we acknowledge inherent limitations in our approach. Our research primarily focused on analyzing the speech-based output of screen readers, neglecting a comprehensive exploration of braille or other critical accessibility devices, which are also used by visually impaired individuals. However, it is important to highlight that the use of speech-based output in screen readers alleviates the need for specific hardware like braille displays for visually impaired users. Our analysis was thorough and insightful in this regard.

**Effect of Faster Speech Rates of Screen Readers.** In our evaluation of screen reader comprehensibility, we focused primarily on the default speech rate of 50%. Given some users may use faster speech rates [61], we briefly also assessed comprehensibility at various faster speech rates using VoiceOver, a smartphone-based screen reader with Authy Authenticator's authentication workflow, including 60%, 70%, 80%, 90%, and 100% speech rates. Our findings indicate that the best comprehensibility score is achieved at the default speech rate, which is 37.32%. Notably, as the speech rate increases, the comprehensibility score drops: 18.06% at 60% speech rates, 1.85% at 70% speech rates, and 0% at speech rates of 80% and above. This implies that authentication flow accessibility and security when using faster rates may significantly degrade.

**Techniques to Mitigate Risks.** Our study has successfully pinpointed several security vulnerabilities that can be actively mitigated through strategic actions by screen reader developers and accessible security researchers. One effective approach to enhance user safety is the integration of automatic phishing detection and malicious link prediction features directly into screen readers. Despite the initial perception of biometric authentication as an easy and secure solution, it presents various challenges for fingerprint [69], face [76], and iris [55], especially for visually impaired individuals who encounter issues in accurately scanning and positioning their fingerprint and face [32]. Screen readers should also detect and inform users about concurrent logins and notification fatigue utilizing notifications received through their devices, and emphasizing the clear communication of service names without overly avoiding specifics can further contribute to preventing cross-service. Resolving conflicts between phone calls and screen readers is essential for improving authentication accessibility for visually impaired users.

**Recommendation and Guidelines.** We recommend that authentication system designers carefully review their authentication workflows and incorporate clear/concise written instructions at each step, ensuring compatibility with screen readers, following our methodological approach (Section 4.2). The designers should manage authentication time expiration to allow visually impaired individuals sufficient time to listen and perform the required actions while preserving usability for sighted users. Additionally, designers should be mindful of potential conflicts with screen reader communication, such as generating a phone call while the screen reader is reading instructions. It is advised to redesign any step that may cause a conflict with screen reader communication. On the other hand, screen readers need improvement in effectively identifying crucial visual elements, such as service names. They should also clearly read out the domain name before loading each page and promptly identify any overlays on the screen. Implementing intelligent security solutions like phishing URL detectors (e.g., warning users if it reads any blacklisted URL) and multiple notification detectors (e.g., warning users if the screen reader detects multiple similar notifications received in the phone's notification bar) in the screen reader can effectively prevent phishing and notification fatigue.

## 7 Conclusion

In this research, we thoroughly examined currently deployed two-factor and passwordless authentication systems, revealing significant vulnerabilities and accessibility challenges, when employed by blind and visually impaired users. Our focus was understanding authentication challenges faced by blind and visually impaired users. Existing methods fall short in balancing accessibility and security for this user group. To comprehensively address these challenges, we advocate cross-disciplinary collaboration involving security, disability services, and human-computer interaction experts. Only through such efforts can we develop authentication methods that are both secure and accessible, enabling equitable digital participation while securing privacy and security.

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

## 8 Appendix

### 8.1 Other Related Works

**Security Concerns for Visually Impaired Users.** Researchers have already noted several security concerns among blind and visually impaired users. Ahmed et al. [1] conducted a study with 14 participants to explore privacy concerns in this group and found that assistive technologies, such as screen readers, raised security concerns due to the risk of bystanders listening to sensitive information. Lazer et al. [58] discussed privacy issues faced by visually impaired individuals when using mobile devices. In another study, Buzzi et al. [22] identified significant accessibility and usability issues for blind users accessing e-commerce platforms via screen readers. Jahankhani et al. [49] examined e-accessibility and security challenges in online banking for blind and visually impaired individuals. Additionally, Napoli et al. [67] investigated measures taken by blind and visually impaired users to ensure their online security and privacy. Their analysis revealed usability issues such as misleading screen reader outputs, and inadequate security advice. These issues can exacerbate the security and privacy risks faced by users. However, their research

does not address concerns regarding authentication technology and its impact on security and accessibility.

**Accessible Authentication Methods.** Several dedicated authentication techniques have been introduced by researchers to facilitate secure authentication for blind and visually impaired users. Haque et al. [41] proposed a gait-based authentication method, utilizing accelerometer sensor data from smartphones. Alnfiai and Sampalli [5] developed BraillePassword, an accessible and observation attack-resistant web authentication application, employing a BrailleEnter keyboard to input characters in the authentication process. Faustino and Girouard [19] devised BendyPass, a password system based on simple bend gestures using a BendyPass device. Kamarushi et al. [54] introduced an authentication technology called OneButtonPIN, utilizing a single interface with on-screen buttons. Ho et al. [45] presented BlindLogin, a dedicated graphical password-based method. However, it should be noted that these methods require special hardware or the implementation of dedicated authentication interfaces to accommodate the unique needs of blind and visually impaired individuals, making it challenging to use in day-to-day activities. Hence, our research emphasizes real-life authentication systems that do not require any special hardware for authentication.

## 8.2 Attack Details

**Threat 0: Remotely Fingerprinting Visually Impaired Users (Detailed).** Attackers are likely to target blind and visually impaired users on the internet, viewing them as potentially vulnerable to different attacks. From the attacker's perspective, detecting blind and visually impaired users could be the prerequisite to exploring further vulnerabilities. Hence, before observing subsequent vulnerabilities in this section, we have observed the possibility of detecting blind and visually impaired users remotely. Momotaz et al. [66] conducted a study with 14 blind users and analyzed 2,000 online posts from three extension-related forums. Their findings indicate that screen reader users rely on extensions to enhance screen reader and application software usability, make partially accessible applications accessible, and receive custom auditory feedback. Numerous extensions are utilized to aid in reading tables, charts, bars, and graphs [23, 31, 80, 89]. ChromeVox, a browser-based screen reader, is itself an extension. Interestingly, these extensions can be identified through browser fingerprinting, an area that has seen extensive research efforts [77, 85, 86]. An attacker may obtain a list of extensions and plugins installed in a victim's browser using browser fingerprinting techniques. By analyzing a victim's installed extensions, an attacker could potentially recognize a visually impaired user and may launch attacks or explore vulnerabilities to compromise credentials.

Sjösten et al. [83] introduced an easy and effective method to collect extension lists based on their web accessibility features. Extensions require web-accessible resources like HTML and JavaScript files to communicate with users through pop-ups or other means. These resources can be accessed directly for Chrome via a schema like *"chrome-extension://extensionid /pathToFile "* and for Mozilla via *"moz-extension://extensionid /pathToFile "*, where *extensionid* is unique for each extension. These resources are exploited to identify extensions, and a positive response from *XMLHttpRequest* indicates that the extension is installed. We conducted testing by installing extensions on Chrome and utilized a third-party website [21] designed to detect extensions, validating their research approach. Our

investigation confirmed the accurate detection of ChromeVox and other listed extensions.

## 8.3 Additional Figures and Tables

**Table 4: Chosen screen readers for various platforms.**

| Platforms | Selected Screen readers |
|---|---|
| Terminal (PC) | JAWS |
| | NVDA |
| | Dolphin |
| | ChromeVox |
| Smartphone | VoiceOver |
| | Talkback |

**Table 5: Authentication methods selected by type.**

| Category | Selected Authentication Methods |
|---|---|
| One Time Password (OTP-2FA) | Google Text Message (Smartphone) |
| | Google Phone Call (Smartphone) |
| | Duo Text Message (Smartphone) |
| | Authenticators: Google authenticator (PC and Smartphone), Microsoft Authenticator (PC and Smartphone), Twilio Authy Authenticator (PC and smartphone), WinAuth Authenticator (PC), GAuth Authenticator (PC(extension)), Authenticator (PC(extension)) |
| Push-2FA | Duo Push (Smartphone) |
| | Google Push (Smartphone) |
| | Microsoft select-confirm Push (Smartphone) |
| FIDO-MFA | Google Titan Security Key (PC and smartphone) |
| Phone Call-2FA | Duo Phone Call (Smartphone) |

**Table 6: Popularity of selected authentication methods.**

| Name | Number of Users/Downloads/Ratings |
|---|---|
| Duo | iPhone: around 1 million ratings and ranked 6 [28] |
| | Android: 10 million downloads [29] |
| Google Authenticator | iPhone: 579.3k ratings and ranked 3 [38] |
| | Android: 100 million downloads [39] |
| Microsoft Authenticator | iPhone: 343.6k ratings and ranked 4 [64] |
| | Android: More than 100 million downloads [65] |
| Twilio Authy Authenticator | iPhone: 38.9k ratings [11] |
| | Android: More than 10 million downloads |
| GAuth Authenticator | Extension: 100000 users [35] |
| Authenticator | Extension: 5000000 users [10] |

**Table 11: Threat analysis for screen reader assisted FIDO-MFA.**

| Threat to FIDO | JAWS | NVDA | Dolphin | ChromeVox | VoiceOver | Talkback |
|---|---|---|---|---|---|---|
| Display Overlay Attack | ● | ○ | ● | ● | | |
| Phishing Attack | ● | ● | ● | ● | ● | ● |
| Malicious Application | ● | ○ | ● | ● | ● | ○ |
| Mis-Authentication / Mis-Registration | ● | ● | ● | ● | ● | |
| Downgrading Attack / Cross-service Attack | ● | ● | ● | ● | ● | ● |

● Susceptible to attack.  ○ Not susceptible.  Empty indicate not tested.

**Table 7: Platforms tested with various two-factor and multi-factor authentication methods.**

| Platform with screen readers (settings) | Authentication Methods |
|---|---|
| Terminal (PC) | One time password (OTP-2FA): Google authenticator (extension), Authy, WinAuth, GAuth (extension) |
| | FIDO-MFA (Titan Security Key) |
| Smartphone | Push-2FA: Google push, Duo push, Microsoft select-confirm |
| | One time password (OTP-2FA): Google (text message, phone call, and authenticator), Duo text message, Microsoft authenticator, Twilo Authy |
| | Duo phone call-2FA |
| | FIDO-MFA (Titan Security Key) |
| Terminal (PC) and Smartphone | Push-2FA: Google push, Duo push, Microsoft select-confirm |
| | One time password (OTP-2FA): Google (text message, phone call, and authenticator), Duo text message, Twilo Authy, GAuth (extension), WinAuth |
| | Duo phone call-2FA |
| | FIDO-MFA (Titan Security Key) |

**Table 8: Devices and platforms used in our evaluation.**

| Device with platform | Screen readers |
|---|---|
| Terminal (PC) with Windows 10 | JAWS |
| | NVDA |
| | Dolphin |
| | ChromeVox (Google Chrome (version107.0.5304.107)) |
| iPhone 7 Plus (iOS) | VoiceOver |
| Samsung Galaxy S21 Ultra (Android) | Talkback |

**Table 9: Comprehensibility of screen readers for general article (non-authentication context).**

| Platform | Selected screen readers | Comprehensibility |
|---|---|---|
| **Terminal (PC)** | JAWS | 79.53% |
| | NVDA | 89.83% |
| | Dolphin | 74.63% |
| | ChromeVox | 87.06% |
| **Mobile** | VoiceOver | 84.74% |
| | Talkback | 84.83% |

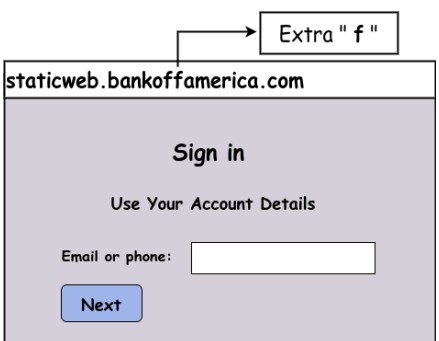

**Figure 2: Phishing link in screen reader assisted scenario.**

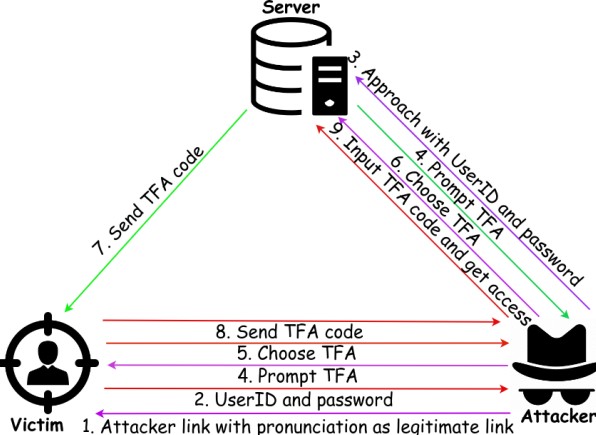

**Figure 3: Potential phishing attacks in authentication methods for screen reader assisted users.**

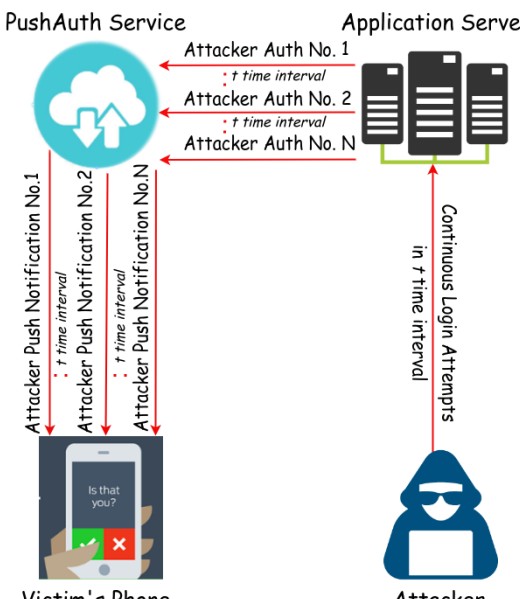

**Figure 4: Susceptibility to fatigue attacks on push-2FA for screen reader assisted users.**

**Table 10: Usability of authentication methods for screen reader-assisted blind and visually impaired users.**

| | | Push-2FA | | | | One Time Password (OTP-2FA) | | | | | | | |
|---|---|---|---|---|---|---|---|---|---|---|---|---|---|
| | | Microsoft, select-confirm | Duo, Push | Google, Push | Duo, call me | Duo, text message | Google, text message | Google, call | GAuth Authenticator | WinAuth Authenticator | Authy Authenticator | Google Authenticator | Google, FIDO |
| **JAWS** | Feasibility | | | | | | | | ● | ○ | ◐ | ● | ● |
| | Exceed Verification Time | | | | | | | | ◐ | ● | ◐ | ◐ | ● |
| **NVDA** | Feasibility | | | | | | | | ● | ○ | ◐ | ● | ● |
| | Exceed Verification Time | | | | | | | | ◐ | ● | ◐ | ◐ | ● |
| **Dolphin** | Feasibility | | | | | | | | ● | ○ | ◐ | ● | ● |
| | Exceed Verification Time | | | | | | | | ◐ | ● | ◐ | ◐ | ● |
| **ChromeVox** | Feasibility | | | | | | | | ● | ○ | ○ | ● | ● |
| | Exceed Verification Time | | | | | | | | ◐ | ● | ● | ◐ | ● |
| **VoiceOver** | Feasibility | ● | ● | ● | ○ | ◐ | ◐ | ○ | | | ● | ◐ | ● |
| | Exceed Verification Time | ◐ | ○ | ○ | ● | ◐ | ◐ | ● | | | ◐ | ◐ | ○ |
| | Conflict Between Instructions | ◐ | ○ | ○ | ● | ○ | ○ | ○ | ○ | ○ | ○ | ○ | ○ |
| **Talkback** | Feasibility | ● | ● | ● | ◐ | ◐ | ◐ | ○ | | | ◐ | ● | ◐ |
| | Exceed Verification Time | ◐ | ● | ○ | ● | ◐ | ◐ | ● | | | ◐ | ◐ | ○ |
| | Conflict Between Instructions | ◐ | ○ | ○ | ● | ○ | ○ | ● | ○ | ○ | ○ | ○ | ○ |
| **JAWS with VoiceOver** | Feasibility | ● | ● | ● | ○ | ◐ | ◐ | ◐ | ● | ◐ | ● | ● | ● |
| | Exceed Verification Time | ◐ | ○ | ○ | ● | ◐ | ◐ | ◐ | ◐ | ● | ◐ | ◐ | ● |
| | Conflict Between Instructions | ◐ | ○ | ○ | ● | ○ | ○ | ○ | ○ | ○ | ○ | ○ | ○ |
| **NVDA with Talkback** | Feasibility | ● | ● | ● | ◐ | ◐ | ◐ | ○ | ◐ | ● | ● | ● | ● |
| | Exceed Verification Time | ◐ | ○ | ○ | ● | ◐ | ◐ | ◐ | ◐ | ● | ◐ | ◐ | ● |
| | Conflict Between Instructions | ◐ | ○ | ○ | ● | ○ | ○ | ● | ○ | ○ | ○ | ○ | ○ |

● indicates that a particular feature is offered by the authentication method.  ◐ indicates a fifty-fifty possibility.
○ indicates that a specific feature is not offered by the method.  Empty means the feature is not relevant or was not tested with the method in this study.

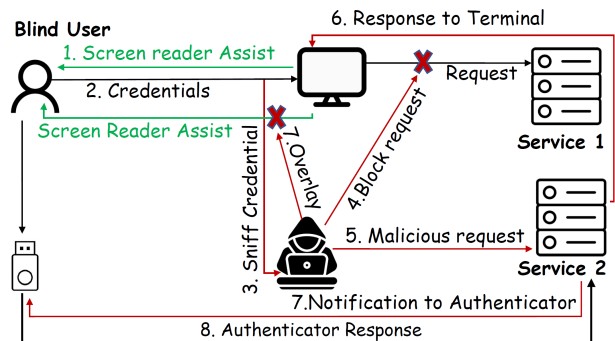

**Figure 5: Scenario of cross-service attack against screen reader assisted FIDO-MFA.**

**Figure 6: Potential shoulder surfing against OTP-2FA for screen reader assisted user.**

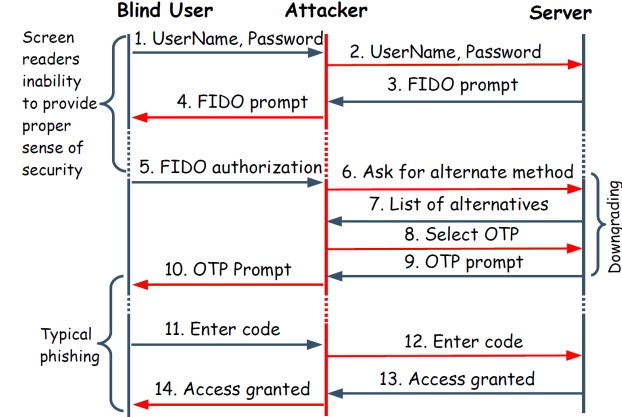

**Figure 7: Susceptibility of downgrading attack against FIDO-MFA for blind user.**

