# OpenReview forum: "Broken Access: On the Challenges of Screen Reader Assisted Two-Factor and Passwordless Authentication"
_ACM.org/TheWebConf/2025/Conference — WWW 2025 Oral_

### Official Review · Reviewer_Hr9H · 2024-11-27

**Novelty:** 5
**Technical Quality:** 4

**Review:**

The paper introduces a framework and analysis for screen reader-assisted authentication, simulating real-world scenarios faced by blind and visually impaired individuals. This is a very important topic for the web community, especially in the areas of security and privacy. The study does a good job of showing the problems and limitations faced by individuals using these services to authenticate, particularly how these issues make them easier targets for attackers who want to exploit their vulnerabilities.

The authors explain that their framework can support future work, helping researchers and developers create better solutions that address the problems identified in this paper. However, it is unclear whether this framework will be made available as open-source for the community to use and test. I recommend the authors make the framework accessible and include this information in the paper, with a link in the final version if the paper is accepted.

There are two additional points from a user’s perspective that could help readers and other stakeholders take action. First, while the authors suggest ways to fix the issues, these ideas are mostly aimed at developers (e.g., improving screen readers or online services). The paper does not provide specific suggestions for users who currently rely on these tools. Including a section that outlines practical steps users can take to reduce security and privacy risks would be very helpful. It could also suggest which solutions might be better suited for different types of users, making the advice more actionable.

Second, it would be useful to provide more details about how many users are affected by this problem to help readers and stakeholders understand the scale of the issue and prioritize accordingly. For example, in Table 3, which combinations of screen readers and authentication methods are used the most? Knowing which combinations are most common could help focus efforts on fixing issues that affect the largest number of users.

**Questions:**

Will the framework be made open-source or otherwise accessible to the broader research and development community? If so, when and where will it be available? If not, what are the authors' reasons for keeping it restricted?

**Reviewer Confidence:**

3: The reviewer is confident but not certain that the evaluation is correct

**Scope:**

3: The work is somewhat relevant to the Web and to the track, and is of narrow interest to a sub-community

---

### Official Review · Reviewer_oQCK · 2024-11-29

**Novelty:** 5
**Technical Quality:** 6

**Review:**

# Summary

The paper studies using two-factor and passwordless authentication mechanisms with screen readers, as employed by blind or visually impaired users. It introduces an assessment framework, called Authentication Workflos Accessibility Review and Evaluation (AWARE), aimed to semi-automatically support the analysis of authentication mechanisms in the screen-reader setting, prior to regular user studies. Using AWARE, the paper systematically assesses a range of PC- and smartphone-based screen readers, when used with a range of one-time password, push-, or token-based two-factor authentication systems. The paper categorizes threads that are specific to or exacerbated by the screen-reader setting and visually impaired users. The findings assess both communicability (which elements can a screen reader express) and comprehensibility (how much of the instructions can a screen reader express), and concluding vulnerabilities by mapping these to the thread categories. The paper concludes with insights and recommendations for both screen readers and the design of authentication system.


# Pros

 + addresses a valuable research question in support of a more inclusive web (authentication) environment
 + semi-automatic approach of AWARE seems reasonable, and qualified to ease evaluation of authentication mechanisms prior to user studies
 + reasonably broad set of authentication mechanisms and screen readers studied


# Cons

 - delineation of what are insights from this work, and what are insights of prior works merely integrated in AWARE is blurry at times (esp. Section 4.3)
 - findings discussion could benefit from a more high-level summary, in turn maybe shortening the detailed discussion of Table 3


# Evaluation

The paper studies a well-motivated setting and fills a gap with supporting semi-automatic analysis of the interplay of authentication mechanisms with screen readers. The covered mechanisms and screen readers show the reasonably wide applicability of the new framework, and interesting conclusions are drawn. Overall, this paper contributes to a more inclusive web (authentication) environment.

**Questions:**

Can you identify the top vulnerabilities/issues you identified with AWARE in your study, and why you deem those the most important ones?

**Reviewer Confidence:**

2: The reviewer is willing to defend the evaluation, but it is likely that the reviewer did not understand parts of the paper

**Scope:**

4: The work is relevant to the Web and to the track, and is of broad interest to the community

---

### Official Review · Reviewer_QfgP · 2024-12-02

**Novelty:** 5
**Technical Quality:** 5

**Review:**

**Summary**

This paper proposes the Authentication Workflows Accessibility Review and Evaluation (AWARE) framework, a semi-automated framework designed for assessing various authentication methods for security and accessibility issues, specifically for blind and visually impaired users. The paper explores the critical challenges, with regards to security, that visually impaired users face in authentication systems, particularly in systems that employ 2FA and passwordless methods and often require the use of multiple devices (e.g., employing OTPs, push notifications, and FIDO-based passwordless methods). The framework provides a comprehensive methodology to evaluate interactions across different screen readers and workflows on both desktop and smartphones, identifying their limitations and potential security vulnerabilities and attacks they are susceptible to, thus offering developers a mechanism to identify and address such problems at an early stage, before involving users for testing. Finally, the authors provide actionable recommendations to developers, such as incorporating clear, concise instructions compatible with screen readers, managing timeouts to accommodate visually impaired users, as well as enhancing screen readers to identify service names, phishing URLs, concurrent logins and attempts of notification fatigue.

**Detailed comments**

- The paper is well-written, easy to follow and understand. It’s an interesting read; I really enjoyed reading this paper.

- It explores a significant real-world problem that is largely unexplored so far: how to make modern web authentication services secure and usable for blind and visually impaired users. This focus on the issues of security and accessibility for a demographic that often faces challenges with existing systems makes this work particularly relevant and impactful.

- The study is comprehensive, examining a wide variety of real-world authentication methods (e.g., OTPs, push notifications, and FIDO systems) across multiple screen readers and platforms (PCs and smartphones), and exploring various attack scenarios.

- This work provides clear and practical recommendations for improving authentication workflows, as well as a framework that can be used by website developers to iteratively assess and improve their systems before involving human testers or deploying them in production. It also provides solid suggestions on how to improve screen reader functionality to mitigate most of the attacks discussed in the paper.

- The framework combines manual setup and automated evaluation using tools like the IBM Watson speech-to-text engine for assessing screen reader output. However, the paper doesn't provide much technical detail on the design, implementation, and assessment of the framework itself. For example, are there errors in the speech-to-text conversion? If so, how are they handled?

- It is also unclear whether the authors evaluated their framework on mockup systems or existing websites. More details on this would strengthen the paper’s methodology. Similarly, while the study includes attack scenarios like phishing and notification fatigue, the implementation details are sparse. Were these attacks simulated in controlled experiments, and if so, how were they designed and executed?

- The paper does not provide enough technical details about how the experiments were set up and run. For instance, it does not specify the experimental design, the number of testers, their expertise, or how consistent the testing procedures were. Additionally, it remains unclear how many runs were performed for each experiment.

- Related to the above, it is also not clear whether visually impaired users were involved in this study as human testers. Moreover, the paper does not specify whether a single tester or multiple testers were used to evaluate the screen-reader-assisted workflows. This point seems important, as individual familiarity with screen readers and authentication workflows could influence the results.

**Questions:**

See comment above about technical details of the framework, as well as details for the experimental design, setup and runs.

**Reviewer Confidence:**

3: The reviewer is confident but not certain that the evaluation is correct

**Scope:**

4: The work is relevant to the Web and to the track, and is of broad interest to the community

---

### Official Review · Reviewer_VU5m · 2024-12-02

**Novelty:** 3
**Technical Quality:** 3

**Review:**

This paper addresses the critical gap in the security and accessibility of web authentication systems for visually impaired users, focusing on screen reader-assisted authentication methods. The authors introduce the so-called AWARE framework, a semi-automated tool for evaluating the security and accessibility of various two-factor and passwordless authentication workflows. They identify significant vulnerabilities and accessibility challenges across multiple scenarios, including phishing, concurrent login, and notification fatigue. The findings highlight the need for improved authentication design that accounts for visually impaired users' unique requirements. The methodology is sound and supported by robust data analysis. However, the absence of user studies and some technical implementation limitations slightly diminish its impact.

**Questions:**

1. Can the AWARE framework be extended to accommodate other accessibility devices like Braille displays?

2. Are there plans to enhance the automation level of AWARE to reduce manual intervention and expand its testing capabilities?

3. While experimental results are compelling, how might findings change when incorporating qualitative feedback from visually impaired users in real-world settings?

**Reviewer Confidence:**

3: The reviewer is confident but not certain that the evaluation is correct

**Scope:**

3: The work is somewhat relevant to the Web and to the track, and is of narrow interest to a sub-community